# Feasibility of Non-Invasive Coronary Artery Disease Screening with Coronary CT Angiography before Transcatheter Aortic Valve Implantation

**DOI:** 10.3390/jcm12062285

**Published:** 2023-03-15

**Authors:** Jérémy Boyer, Axel Bartoli, Pierre Deharo, Antoine Vaillier, Jérôme Ferrara, Pierre-Antoine Barral, Nicolas Jaussaud, Pierre Morera, Alizée Porto, Frédéric Collart, Alexis Jacquier, Thomas Cuisset

**Affiliations:** 1Département de Cardiologie, CHU Timone, 13005 Marseille, France; 2Department of Radiology, CHU Timone, AP-HM, 264, Rue Saint-Pierre, 13005 Marseille, France; 3CRMBM-UMR CNRS 7339, Aix-Marseille Université, 27, Boulevard Jean Moulin, CEDEX 05, 13385 Marseille, France; 4Aix-Marseille Université, Inserm, Inra, C2VN, 13005 Marseille, France; 5Faculté de Médecine, Aix-Marseille Université, 13005 Marseille, France; 6Service de Chirurgie Cardiaque, CHU Timone, 13005 Marseille, France

**Keywords:** computed tomography angiography, transcatheter aortic valve implantation, percutaneous coronary intervention

## Abstract

Coronary artery disease (CAD) screening is usually performed before transcatheter aortic valve implantation (TAVI) by invasive coronary angiography (ICA). Computed coronary tomography angiography (CCTA) has shown good diagnostic performance for CAD screening in patients with a low probability of CAD and is systematically performed before TAVI. CCTA could be an efficient alternative to ICA for CAD screening before TAVI. We sought to investigate the diagnostic performance of CCTA in a population of unselected patients without known CAD who were candidates for TAVI. All consecutive patients referred to our center for TAVI without known CAD were enrolled. All patients underwent CCTA and ICA, which were considered the gold standard. A statistical analysis of the diagnostic performance per patient and per artery was performed. 307 consecutive patients were enrolled. CCTA was non-analyzable in 25 patients (8.9%). In the per-patient analysis, CCTA had a sensitivity of 89.6%, a specificity of 90.2%, a positive predictive value of 65.15%, and a negative predictive value of 97.7%. Only five patients were classified as false negatives on the CCTA. Despite some limitations of the study, CCTA seems reliable for CAD screening in patients without known CAD who are candidates for TAVI. By using CCTA, ICA could be avoided in patients with a CAD-RADS score ≤ 2, which represents 74.8% of patients.

## 1. Introduction

Transcatheter aortic valve implantation (TAVI) is the standard treatment for symptomatic severe aortic stenosis in patients at high surgical risk [1]. In this population, the prevalence of coronary artery disease (CAD) is high, ranging between 60 and 75% [2,3]. Current guidelines recommend a pre-procedural screening for CAD in the management of valvular heart disease and TAVI, in particular [1]. Invasive coronary angiography (ICA) remains the most frequently used method for CAD screening in patients with severe aortic stenosis and a high clinical likelihood of CAD. However, TAVI indications have recently been extended to lower surgical-risk patients, in whom the prevalence of CAD tends to be lower [4]. Coronary computed tomography angiography (CCTA) is now considered an alternative to ICA before valve intervention in patients with a low probability of CAD [1]. It has shown high sensitivity and negative predictive value for CAD diagnoses, allowing the exclusion of CAD with a high confidence level, and is recommended as an initial assessment for patients with a low clinical likelihood of CAD [5,6]. CCTA performance rates in patients with stented coronary arteries were lower [7].

The wider indications for TAVI, the systematic need for a pre-procedural injected CT for aortic valve anatomy assessment, and the diagnostic performance of CCTA in CAD suggest that CCTA could reduce the need for ICA in pre-procedural patient management. This could lead to faster and more ambulatory pre-procedural screenings, lower doses of iodine contrast medium, and lower costs. Several studies have attempted to evaluate the ability of CCTA to detect significant CAD in the pre-TAVI work-up, with varying results [8,9]. Two recent meta-analyses showed that CCTA had excellent diagnostic accuracy for assessing obstructive CAD in patients referred for TAVI [10,11]. But there is still a need for recent supplementary data, especially at the vessel level.

The main objective of this study is to evaluate the diagnostic accuracy of CCTA for the detection of significant CAD in patients undergoing TAVI.

## 2. Materials and Methods

### 2.1. Study Design and Population

This single-center (CHU Timone, AP-HM, Marseille, France) retrospective study was conducted from March 2020 to March 2022, complied with the Declaration of Helsinki, and was approved by the local ethics committee. All patients had signed and written informed consent. All consecutive adult patients referred to our center for TAVI for severe symptomatic aortic stenosis were enrolled, and data were collected in our database (which is part of the France TAVI registry) [12]. Given our daily clinical practice and based on recommendations, all patients underwent a pre-interventional CCTA to assess valve morphology and an ICA for CAD evaluation. Patients with a prior history of PCI or coronary artery bypass grafting (CABG) were not included. Patients with incomplete CCTA or who underwent PCI before CCTA were excluded. All clinical characteristics, cardiovascular risk factors, cardiac and aortic valve parameters, CCTA measurements, and PCI data were recorded. A flow chart is presented in Figure 1.

### 2.2. CCTA Parameters and Analysis

CCTA was performed using a 256-row detector CT (Revolution Frontier; General Electric Healthcare, Wauwatosa, WI, USA) with prospective cardiac gating and the following acquisition parameters: Gantry rotation: 0.28 s; tube current: 200–500 mA; tube voltage: 100–120 kVp; slice thickness: 0.625 mm; pitch: 1.375; field-of-view: 18 cm. The tube current and voltage were adapted to the patient’s BMI. An intravenous injection of 70–100 mL of iohexol (350 mg I/mL, Omnipaque^®^ 350, GE Healthcare) was performed in a brachial vein using a power injector followed by a saline chaser in a triphasic injection protocol [13]. Images were acquired after the bolus tracking technique. Heart rate control medication was provided if the heart rate was greater than 80 bpm.

The CCTA analysis was conducted by an experienced radiologist (A.B., 5 years of experience) who was blinded to patient characteristics and ICA results. The four main coronary arteries, left main stem (LMS), left anterior descending artery (LAD), left circumflex artery (LCx), and right coronary artery (RCA), were evaluated according to the CAD-RADS classification score (CAD-RADS 0: no stenosis; 1: 1–24% stenosis; 2: 25–49% stenosis; 3: 50–69% stenosis; 4: 70–99% stenosis or left main stem stenosis ≥50%; 5: totally occluded artery) [14]. If multiple stenoses were found in a single vessel, the most severe one defined the CAD-RADS artery score. A global CAD-RADS score for each patient was assigned according to the highest CAD-RADS score of the four main arteries. Image quality was assessed and documented per artery on a four-point scale (0–3) presented in Figure 2. The 0 scale: non-diagnostic image quality, meaning significant coronary stenosis cannot safely be ruled out; 1: fair image quality; 2: good image quality, minor artefacts; 3: excellent image quality, absence of artefacts, very good delineation of the coronary lumen. The global quality score retained for each patient was the lowest of the four main arteries we analyzed.

### 2.3. Invasive Coronary Angiography Analysis

All patients underwent an ICA prior to TAVI. All ICA were reviewed by an independent committee composed of two interventional cardiologists from APHM; in case of discordance between the cardiologists, a third operator was requested. Significant stenosis was defined by an artery diameter reduction ≥50%. At the vessel level, vessels with one or more coronary segments considered to have significant stenosis were defined as positive. If more than one lesion was present within the same vessel, the most severe lesion was selected as representative of the vessel.

### 2.4. Outcomes

The main outcome was to evaluate the diagnostic accuracy of CCTA for pre-procedural screening of CAD in patients undergoing TAVI in comparison to the gold standard, ICA. The diagnostic performance evaluation was done at both the patient level and for each coronary artery.

For secondary outcomes, we compared PCI incidence between CAD-RADS ≤ 2 and CAD-RADS ≥ 3. We also described non-analyzable patient characteristics.

### 2.5. Statistical Analysis

The diagnostic performance of CCTA was compared with that of ICA by calculating sensitivity (Se), specificity (Sp), positive predictive values (PPV), and negative predictive values (NPV). The same indicators were calculated for the comparison between the diagnostic performance of CCTA and ICA for each artery (LMS, LAD, LCx, RCA). Patient characteristics were first described and compared between the 2 groups of interest (CAD.RADS ≤ 2 vs. CAD-RADS ≥ 3; analyzable CCTA vs. non-analyzable CCTA). Quantitative variables were presented as mean and standard deviation in cases of normal distribution (the assumption of normal distribution was assessed graphically using histograms and Q-Q plots), otherwise as median and interquartile ranges, and compared using the Student’s t-test if valid (and using the Mann-Whitney U test if otherwise). Categorical variables were presented as numbers (percentages) and compared using the chi-squared test if valid (and Fisher’s exact test if otherwise). All tests were two-tailed, and *p* values < 0.05 were considered statistically significant. Statistical analyses were performed using RStudio software Version 2022.07.1 (RStudio, Boston, MA, USA).

## 3. Results

### 3.1. Population Characteristics

During the study period, 651 patients underwent TAVI. Overall, 286 patients (44%) were ineligible because of a known history of CAD, TAVI for aortic regurgitation, or bioprosthetic valve deterioration. Among the remaining 365 patients, 58 (17%) were excluded due to unavailable CCTA (81%), incomplete CCTA (7%), and PCI performed before realization of CCTA (12%). A total of 307 patients (47.2% of all TAVI performed during the study period) were included for analysis. Among these, 25 (8.1%) were not analyzable and were classified as 0 for the global image quality score (major artefacts, insufficient opacification, or coronary artery calcium overload on at least one artery). 282 CCTA (91.9%) were analyzable (global image quality score ≥1). In the analyzable group, the global image quality score was 2.8. A flow chart is presented in Figure 1. Patient characteristics are presented in Table 1. The STS score and history of stroke were significantly higher in patients with non-analyzable CCTA.

### 3.2. CCTA and ICA Analysis

Among all analyzable CCTAs (*n* = 282), no patient had CAD-RADS 0, 69 had CAD-RADS 1, 147 had CAD-RADS 2, 48 had CAD-RADS 3, 17 had CAD-RADS 4, and 1 had CAD-RADS 5 (Figure 3). CCTA results in comparison to ICA are described in Table 2 and Figure 4.

At an artery-level evaluation, the LAD had the most misclassified lesions on the CCTA. False positive lesions were classified as CAD-RADS 3 in 96.22%, and false negative lesions were mainly classified as CAD-RADS 2 (69.0%).

On a patient-level analysis, five patients were considered false negatives. All of them were classified as CAD-RADS 2 on the CCTA, with the highest CAD-RADS measured in the LAD. Two had a CCTA quality score of 2, and the three remaining had a quality score of 1 (mean quality score: 1.4). Among these 5 patients, 4 had significant lesions in the LAD, and one had two lesions, according to the ICA, one in the LAD and one in the RCA. Two underwent PCI to treat lesions of the LAD artery. One had a 60% stenosis with a positive FFR, and the other had a 90% stenosis. The three remaining patients received only medical treatment. In patients with CAD-RADS 3, 4, and 5, there were 23 false positives (34.8%).

According to the ICA, 48 patients had significant coronary lesions, and 17 (35.4%) were treated with PCI. The main site of PCI was the left anterior descending artery (58.8%), followed by the right coronary artery (35.2%) and the left main stem (6%). No PCI was performed in the circumflex artery. The mean CAD-RADS score was 3.3 in patients who underwent PCI.

### 3.3. Diagnostic Performance of CCTA

The diagnostic performance of CCTA for discriminating significant coronary artery lesions is presented in Table 2 and the Graphical Abstract and includes both artery-level and patient-level data. For differentiating patients with significant lesions (CAD-RADS ≥ 3) versus non-significant lesions (CAD-RADS ≤ 2), CCTA had a sensitivity of 89.6% and a negative predictive value of 97.7%. Specificity was 90.1%, and the positive predictive value was relatively low at 65.1%. The CCTA positive likelihood ratio was 9.1 and the negative likelihood ratio was 0.1. NPV was measured at 99.6%, 96.6%, 99.6%, and 97.7% for the LMS, LAD, LCx, and RCA, respectively.

## 4. Discussion

In this observational, monocentric, retrospective study, we showed that in most cases, CCTA is reliable to screen for CAD before TAVI in patients with indications for severe symptomatic aortic stenosis but without known CAD. CCTA appeared to have a high diagnostic performance, allowing the safe exclusion of CAD in this population.

Our study showed that CCTA had a high sensitivity of 89.6% and a negative predictive value of 97.7%, discriminating between patients with CAD-RADS ≤ 2 and patients with CAD-RADS ≥ 3. These results are in line with other studies on the diagnostic performance of CCTA for detecting CAD in the general population [5,6,7,15] and before TAVI [9,11,16,17]. Our population was composed of patients with a low mean surgical risk, as assessed by the EuroSCORE II and the STS score. This was in accordance with the actual profile of patients undergoing TAVI. The characteristics of our study’s population were similar to those in other studies proposing TAVI to low-risk patients, except for the absence of CAD in our cohort [4,18]. We chose to use the CAD-RADS score to describe the coronary lesions demonstrated on the CCTA because of its correlation with the ICA and its prognostic value [14]. Indeed, Xie et al. [19] demonstrated in their registry, which included about 5000 patients, that a higher CAD-RADS score was associated with a higher rate of cardiovascular events at 5 years. In the randomized controlled SCOT-HEART study [20], the use of CCTA and classification of lesions by CAD-RADS score in addition to standard care in patients with stable chest pain resulted in a significantly lower rate of death from coronary heart disease or non-fatal myocardial infarction at 5 years than standard care alone without resulting in a significantly higher rate of ICA. We defined coronary stenosis greater than 50% as significant by analogy to the CAD-RADS score [14]. Indeed, the latter classifies stenoses greater than 50% in CAD-RADS 3, justifying further exploration [14]. In this particular population, we wanted to demonstrate that we could free ourselves from ICA use in the case of a CAD-RADS score ≤ 2. In addition, other studies on the subject have also used a significance level of 50% [8,17,21].

It appears that CCTA is analyzable in more than nine out of ten cases. The percentage of CCTA that is analyzable is slightly higher than in other studies on CCTA performed pre-TAVI [17,21]; however, these studies included patients with CAD, who had more potential artefacts due to the presence of coronary stents. Andreini et al. [9] showed that in a population composed of 325 patients referred for TAVI, including patients with previous CABG and PCI, CCTA was analyzable in 95.6% of cases using a 256-row detector. Advancements in CCTA technology could allow for higher feasibility and include patients with atrial fibrillation or with high heart rates, as shown in a study using a 16-cm-wide computed tomography detector [22]. In our study, the main causes of non-analyzable CCTA were motion artefacts and heavily calcified coronary arteries. It is important to complete the available data on the subject because of the technical improvements that can improve the diagnostic performance of CCTA. In our study, we used a 256-row detector CT, while in the available meta-analyses, most studies used a 64- or 128-row detector CT [10,11].

In our study, 58 eligible patients (17%) were excluded because of unavailable or incomplete CCTA or PCI performed before CCTA. We think that this corresponds to real-world situations. However, other studies have had fewer excluded patients. Gohmann et al. [21] excluded only 7 of 517 patients (1.4%) due to an incorrect CT protocol; this could be explained by the prospective design of the study. In their retrospective study, Strong et al. [8] excluded 30 of 342 patients (8.8%); this could be explained by the fact that they included only patients who had CCTA available to them, as well as patients for whom PCI was performed before CCTA. They excluded patients for whom the time between CCTA and ICA was greater than 6 months.

We analyzed the diagnostic performance of CCTA for each coronary artery. It appears that the negative predictive value was slightly lower for the left anterior descending artery. This result is compatible with other studies using similar computed tomography protocols [17,21]. This could be explained by more motion artefacts affecting this artery because of the long path of the interventricular groove. However, a patient-based analysis seems to be more appropriate for daily practice decisions than the vessel-based analysis.

In our study, a total of 211 patients had a CCTA showing a CAD-RADS score ≤ 2 and no lesion according to ICA. Hence, we can suppose that 74.8% of the ICAs were performed in patients without significant coronary lesions and could have been avoided, which could have a major impact in this elderly population. In a study [23] concerning 491 patients (including patients with a history of CAD) who underwent TAVI, ICA was performed only in patients for whom CCTA was not analyzable or showed significant coronary lesions. The authors showed that using CCTA for CAD screening was safe, with no more MACE after one year in this group. ICA was performed in 24% of patients. In our study, only 48 patients had significant coronary stenosis, 5 of whom were considered negative by the CCTA. PCI was performed in 35.4% of them (2 in false-negative patients). The benefits of PCI before TAVI remained unclear due to a lack of evidence. Several observational studies have suggested that there is no difference in clinical outcomes between patients undergoing TAVI and PCI versus isolated TAVI [24,25,26]. The randomized ACTIVATION trial showed no benefit in terms of death and rehospitalization when comparing patients who had PCI before TAVI and patients without PCI [27]. The latter included symptomatic patients with CCS-2 angina and lesions >70% in arteries >2.5 mm, but left main stem lesions were excluded. Another randomized trial, NOTION 3 (NTC030558627), is underway to evaluate the benefit of complete coronary revascularization (stenosis with FFR ≤0.80 or stenosis 90%) in patients scheduled for TAVI. Together with our data and the previously suggested feasibility of "CCTA-only" screening for CAD before TAVI, this represents a good argument for a more selective use of ICA before TAVI. In addition to a prior history of CAD, other aspects might drive the decision towards ICA before TAVI, such as the presence of angina as a symptom or the severity of AS.

In summary, our single-center observational study suggests that CCTA seems reliable in patients before TAVI for symptomatic aortic stenosis and provides good diagnostic performance that safely excludes CAD. CCTA might be an effective non-invasive CAD screening tool in patients before TAVI without known CAD.

The study has several limitations. First, because of the retrospective interpretation of CCTA, the findings of the present investigation are subject to confounding bias. Second, based on the observational design, the collection of some data could have been incomplete (the missing data were mainly weight and height, as well as some information needed to calculate the STS score). Third, about 50% of patients were excluded, which can lead to a selection bias. Fourth, this study is monocentric, which could limit extrapolation of the results to other populations, and the rate of PCI could also be different with other teams. Fifth, the relatively low number of included arteries could result in a lack of power to study the diagnostic performance of each artery. Sixth, the reading of the CCTA by a single radiologist did not allow for a reproducible inter-observer evaluation.

In a population of patients without known CAD who would benefit from TAVI for symptomatic severe aortic stenosis, CCTA is an easily feasible, non-invasive tool for CAD screening with good diagnostic performance, allowing for the exclusion of CAD with a high degree of confidence. The diagnostic performance for each coronary artery was good. Performing CCTA alone in cases where there is no significant CAD could lead to a reduction in procedural risks, procedural costs, and hospitalization time. However, despite its good diagnostic performance, CCTA alone before TAVI must be investigated in randomized, controlled prospective studies to evaluate its safety in terms of the occurrence of clinical events.

## Figures and Tables

**Figure 1 jcm-12-02285-f001:**
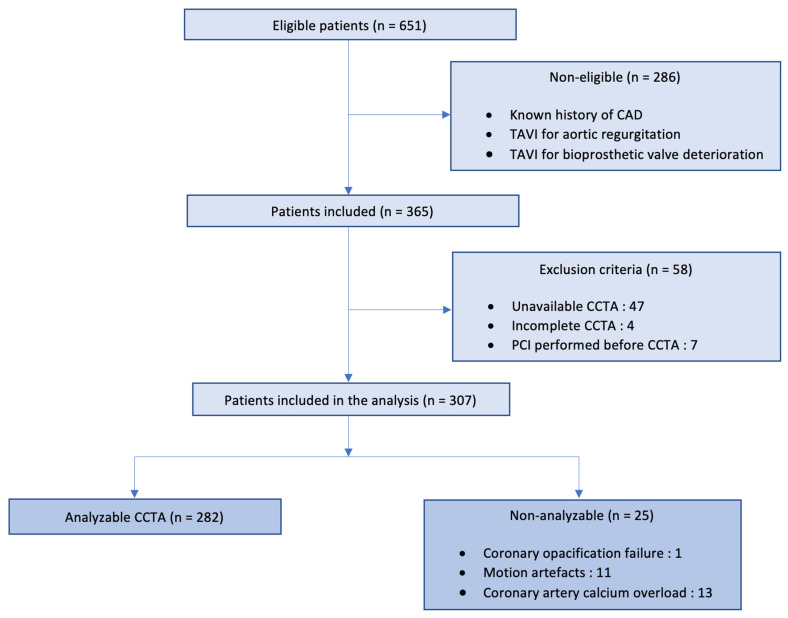
Flow chart diagram.

**Figure 2 jcm-12-02285-f002:**
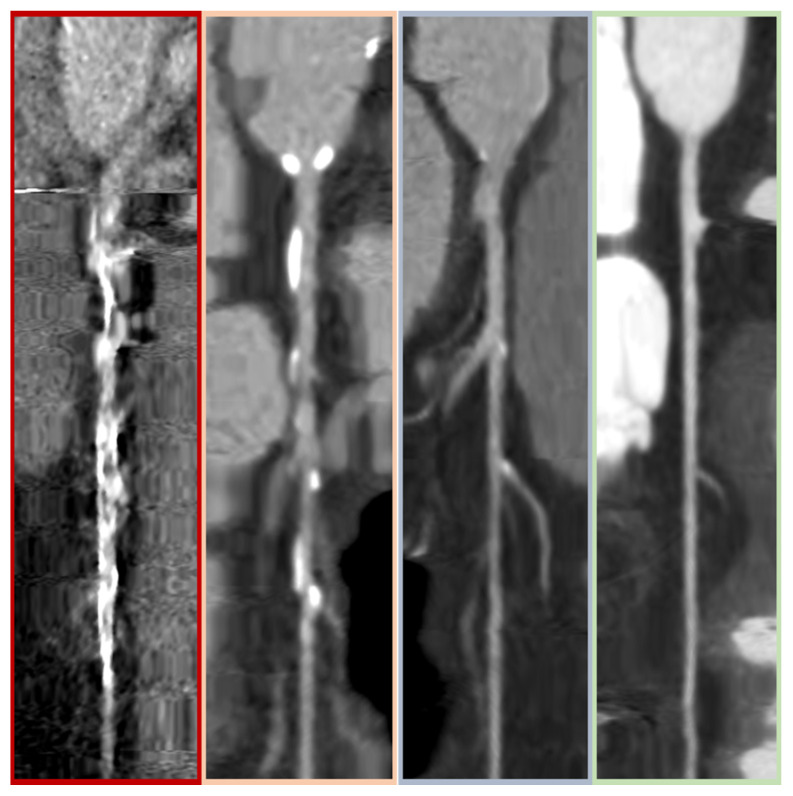
CCTA quality criteria. Examples of CCTA imaging for each quality score category, from left to right, from 0 to 3.

**Figure 3 jcm-12-02285-f003:**
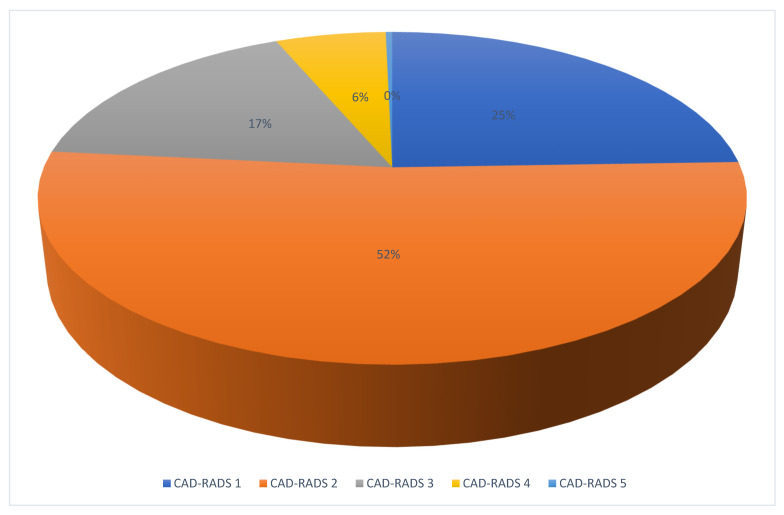
Repartition of patients by CAD-RADS score.

**Figure 4 jcm-12-02285-f004:**
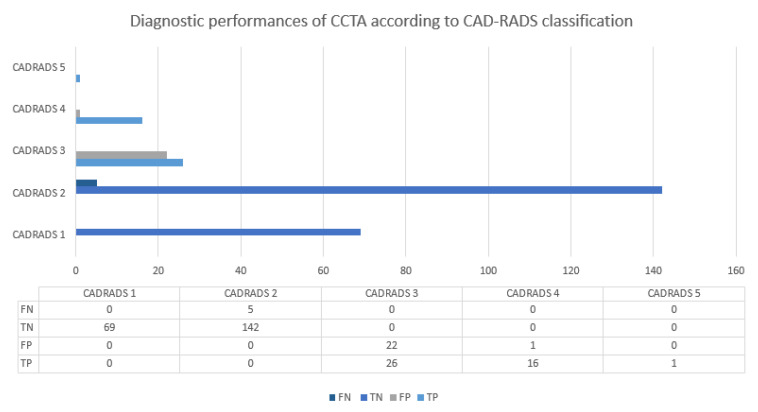
Results of CCTA and ICA according to the CAD-RADS category.

**Table 1 jcm-12-02285-t001:** Population characteristics (*n* = 307).

	Analyzable CCTA (*n* = 282)	Non-Analyzable CCTA (*n* = 25)	*p*
Male sex, *n* (%)	122 (43.3)	12 (48)	0.647
Age (yrs), mean ± SD	82.1 ± 7.2	82.3 ± 7.3	
BMI (kg/m^2^), mean ± SD	26.6 ± 5.05	27.5 ± 5.4	0.250
EuroSCORE II, mean ± SD	3.52 ± 2.65	3.54 ± 2.25	1
STS score, mean ± SD	3.56 ± 3.85	4.16 ± 0.92	*0.019*
Previous medical history			
Diabetes, *n* (%)	81 (28.7)	9 (36)	0.444
Hypertension, *n* (%)	200 (70.9)	20 (80)	0.334
Dyslipidemia, *n* (%)	110 (39)	13 (52)	0.204
Current smokers, *n* (%)	18 (6.4)	1 (4)	1
Atrial fibrillation, *n* (%)	80 (28.4)	4 (16)	0.462
Chronic kidney disease, *n* (%) ^§^	23 (8.2)	3 (12)	0.456
Peripheral vessel disease, *n* (%) ^§§^	15 (5.32)	3 (12)	0.172
Stroke, *n* (%)	20 (7.09)	5 (20)	*0.048*
Amyloidosis, *n* (%)	25 (8.80)	4 (16)	0.275
Active cancer, *n* (%)	104 (36.9)	6 (24)	0.198
Echocardiographic characteristics			
Indexed aortic valve area (cm^2^/m), mean ± SD	0.42 ± 0.11	0.41 ± 0.10	0.853
Mean gradient (mmHg), mean ± SD	53.2 ± 15.9	51.6 ± 11.7	0.770
Peak velocity (m/s), mean ± SD	4.5 ± 0.7	4.6 ± 0.4	0.616
LVEF (%), mean ± SD	59.9 ± 11.8	60.8 ± 9	0.905
Tomography characteristics			
Aortic valve calcium score, mean ± SD	3136.1 ± 1987.3	3494.9 ± 1575.6	0.112
Iodine contrast product (mL), mean ± SD	84.7 ± 9.9	85 ± 10.1	0.790
Dose length product (mGy/cm), mean ± SD	1377.1 ± 553	1756.4 ± 1079.3	0.172

Values are presented as *n* (percentages) or the mean ± standard deviation. ^§^ Defined as creatinine clearance ≤30 mL/min/1.73 m^2^ by the MDRD formula. ^§§^ Included lower extremity artery, upper extremity artery, carotid and vertebral arteries, mesenteric artery, and renal artery diseases. BMI: body mass index; LVEF: left ventricular ejection fraction; CCTA: coronary computed tomography angiography.

**Table 2 jcm-12-02285-t002:** Diagnostic performance of CCTA compared to ICA for analyzable patients (*n* = 282).

	TN	TP	FP	FN	Se	Sp	PPV	NPV
LMS	280	0	1	1	0	99.6	0	99.6%
LAD	224	22	28	8	73.3%	88.9%	44%	96.6%
LCx	266	5	11	1	83.3%	96%	31.3%	99.6%
RCA	250	16	11	6	72.7%	95.8%	59.2%	97.7%
Overall	211	43	23	5	89.59%	90.17%	65.15%	97.69%

LMS: left main stem; LAD: left anterior descending artery; LCx: circumflex artery; RCA: right coronary artery; Se: sensitivity; Sp: specificity NPV: negative predictive value; PPV: positive predictive value.

## Data Availability

No new data were created or analyzed in this study. Data sharing is not applicable to this article.

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
