# Peer review of "Feasibility of Non-Invasive Coronary Artery Disease Screening with Coronary CT Angiography before Transcatheter Aortic Valve Implantation"

_jcm, 2023, doi:10.3390/jcm12062285_

Round 1
Reviewer 1 Report
The report deals with pre TAVI coronary assessment. ICA is considered as gold standard. CCTA in certain scenarios can help or replace ICA before TAVI.
While it is intereseting there ar some issues regarding the included patiets need to be clarified.
In some places English is not the best , I would advise English lecturing.
The only question is which cases can you rely only on data from CCTA.
It is wise to exclude patients with prior PCI, these patients you need ICA anyway, but it would be ideal if the prior treated vessels could be investigated with CCTA. Please comment on this. .
Of the remaining patients there is still a lot of patients lack CCTA data as described in row 143. , "58 (17%) were excluded due to unavailable CCTA (81%), incomplete CCTA (7%) and PCI performed before realization of CCTA 144 (12%)"
It is real world scenario.
By the end of the day 282 of 651 patients remain.
In row 223 it is written that in this study CCTA ad good performance compared to others. Since in tose studies no exclusions were present it is misleading. (less than 50% of TAVI patient data was analyzed here) Please comment on this or describe differences better.
There is still debate on which lseions to treat More data in the discussion would be good. (LM: 50, prox lesions 50, or 70%...)
In the summary part I think CCTA only strategy should be used with more caution due to the described false interpretation of CCTA in cases and high number of non analyzable CCTA data.
In the limitations section maybe there should be more emphasis on current CCTA technology limitations.
Author Response
Reviewer 1
Thank you for your thoughtful review of this study. We have modified the manuscript in accordance with your comments.
In some places English is not the best, I would advise English lecturing.
The revised version of the manuscript was proofread by English medical writer. You will find the final version of the revised paper.
The only question is which cases can you rely only on data from CCTA.
It is wise to exclude patients with prior PCI, these patients you need ICA anyway, but it would be ideal if the prior treated vessels could be investigated with CCTA. Please comment on this.
This would have been interesting, but the aim of our study is to demonstrate in patients at low risk of coronary artery disease that CCTA is a safe alternative to angiography. CCTA in patients with a history of coronary angioplasty was not analyzed. We are unable to provide these data.
It is very likely that in a near future, the improvement in CCTA technology will allow an accurate assessment of stented segments / vessels. Therefore, patients with prior PCI would be potentially analyzed by CCTA. However, at the moment a previous stent is a contra indication or precaution to CCTA analysis.
Of the remaining patients there is still a lot of patients lack CCTA data as described in row 143. , "58 (17%) were excluded due to unavailable CCTA (81%), incomplete CCTA (7%) and PCI performed before realization of CCTA 144 (12%)"
It is real world scenario.
By the end of the day 282 of 651 patients remain.
In row 223 it is written that in this study CCTA ad good performance compared to others. Since in tose studies no exclusions were present it is misleading. (less than 50% of TAVI patient data was analyzed here) Please comment on this or describe differences better.
In our study, 58 eligible patients (17%) were excluded because of unavailable or incomplete CCTA, or PCI performed before CCTA. We think that this corresponds to real-world situations. However, other studies have had fewer excluded patients. Gohmann et al. (20) excluded only 7 of 517 patients (1.4%) due to incorrect CT-protocol; this could be explained by the prospective design of the study. In their retrospective study, Strong et al. (8) excluded 30 of 342 patients (8.8%); this could be explained by the fact that they included only patients who had CCTA available to them, as well as patients for whom PCI was performed before CCTA. They excluded patients for whom the time between CCTA and ICA was greater than 6 months.
There is still debate on which lseions to treat More data in the discussion would be good. (LM: 50, prox lesions 50, or 70%...)
There is still debate on the PCI indications in pre TAVI scenario. Indeed, several observational studies have suggested that there is no difference in clinical outcomes in patients undergoing TAVI and PCI versus isolated TAVI. Moreover, following the ACTIVATION trial the trend is in favor a a very large proportion of medically managed coronary aretery disease in TAVI patients. In the study PCI was associated with increase adverse events, mostly related to bleeding.
In the summary part I think CCTA only strategy should be used with more caution due to the described false interpretation of CCTA in cases and high number of non analyzable CCTA data.
Despite some limitations of the study, CCTA seems reliable for CAD screening in patients without known CAD who are candidates for TAVI. We tempered this in the abstract.
We couldn’t highlight it more in the summary because of a limited number of words authorized. We detailled limitations in the discussion.

Reviewer 2 Report
In this study Authors aimed to evaluate the diagnostic accuracy of CCTA for the detection of significant CAD in patients undergoing TAVI. The subject itself is important, however methods and data presentation must be clarified.
Why, in this study, 50% stenosis was considered significant in each artery, instead of 70%?
Figure 1, 284 were not eligible, instead of analysable I believe,
Ad the aim of the study is to evaluate the diagnostic accuracy of CCTA for the detection of significant CAD in patients undergoing TAVI - I do not understand why Authors divided population into analysable, and non-analysable (Table.1). It is not connected to research question and this distinction and reasoning is not mentioned/ explained anywhere else.
In paragraph “CCTA and ICA analysis” Authors explain that 2 out of 5 patients underwent PCI, however they do not state what was the exact % of stenosis severity.
What is a rationale behind inclusion factor in multivariable analysis? As shown in univariate analysis, most of them were insignificant. What was the methodology of performing multivariate analysis? How exactly the model was constructed?
Discussion
In the discussion, the statement “These results are better than other studies about CCTA in patients before TAVI” sound improper. May Authors explain what they mean?
What Authors mean that the length of LAD is important?
“Makes more sense” is not an official language.
What Authors mean by saying that “Among them, 35.4 % had benefited from PCI”, how the benefit was measured? In the following sentence authors say that “The benefit of PCI before TAVI remained unclear.”. It is a bit counterintuitive.
In limitations Author mention that some data might be incomplete – which data exactly were incomplete? That should be clarified.
Author Response
Reviewer 2
Thank you for your thoughtful review of this study. We have modified the manuscript in accordance with your comments.
Why, in this study, 50% stenosis was considered significant in each artery, instead of 70%?
We defined coronary stenosis greater than 50% as significant by analogy to the CAD-RADS score. Indeed, the latter classifies lesions greater than 50% in category 3 justifying further exploration. In this particular population, we wished to propose a protocol that would allow us to dispense with coronary angiography in case of a CAD-RADS score strictly inferior to 3. In addition, other studies on the subject also use a significance level of 50%. We now specify this in the revised version.
At the moment there is no classification or recommendation on the significance of coronary artery disease when treating a patient pre TAVI. FFR is an option but associated with limitations. The objective of our study was to identify coronary artery disease and therefore we relied on the radiographic CAD-RADS score. ESC guidelines suggest that PCI should be considered in patients with a primary indication to undergo TAVI and coronary artery diameter stenosis >70% in proximal segments (class Iia, level of evidence C).
Figure 1, 284 were not eligible, instead of analysable I believe,
Thanks for spotting this mistake. We corrected it.
Ad the aim of the study is to evaluate the diagnostic accuracy of CCTA for the detection of significant CAD in patients undergoing TAVI - I do not understand why Authors divided population into analysable, and non-analysable (Table.1). It is not connected to research question and this distinction and reasoning is not mentioned/ explained anywhere else.
In this Table 1, we wanted to highlight the characteristics of the patients that could be analyzed first. Nevertheless, it seemed useful to us to put in parallel the characteristics of the non-analyzable patients for purely descriptive purposes, although this is not in fact related to the main objective of our research. This is now specified in the methodology.
In paragraph “CCTA and ICA analysis” Authors explain that 2 out of 5 patients underwent PCI, however they do not state what was the exact % of stenosis severity.
The first patient had a stenosis of 60 % with positive FFR. The second a stenosis of 90 %. We now mentioned it in the manuscript.
What is a rationale behind inclusion factor in multivariable analysis? As shown in univariate analysis, most of them were insignificant. What was the methodology of performing multivariate analysis? How exactly the model was constructed?
Indeed, you are absolutely right. This analysis is not related to the objective of our study which is not large enough to show a relevant result. To avoid any confusion we have removed this part of the manuscript.
In the discussion, the statement “These results are better than other studies about CCTA in patients before TAVI” sound improper. May Authors explain what they mean?
We wanted to say that the percentage of analyzable CCTA in our study was slightly higher than in other similar studies, while emphasizing that the latter included patients with stented CAD. We have reformulated this.
What Authors mean that the length of LAD is important?
We wanted to expose the fact that because of its long course in the interventricular groove, LAD was more likely to be the site of motion artefacts. Moreover, LAD being the main vessel in charge of left ventricle myocardial oxygenation its analysis should be particularly accurate.
“Makes more sense” is not an official language.
Thank you. We modified it. The revised version of the manuscript was proofread by a English medical writer
What Authors mean by saying that “Among them, 35.4 % had benefited from PCI”, how the benefit was measured? In the following sentence authors say that “The benefit of PCI before TAVI remained unclear.”. It is a bit counterintuitive.
Indeed, this seems counter-intuitive. We have modified this part to make it clearer.
In limitations Author mention that some data might be incomplete – which data exactly were incomplete? That should be clarified.
Missing data were mainly weight and height as well as some informations to calculate the STS score. We mention it now.

Reviewer 3 Report
I read with pleasure this article by Boye and colleagues. Here are my comments:
- The retrospective and monocentric design, together with the small sample size, are major limitation.
- The author themselves acknowledge that there are meta-analyses (which means several studies) on the same subject. It is unclear what the study add to the literature.
- Why was ICA adjudicated by two cardiologist whereas coronary CT scan just by one radiologist? CT scan should be of main interest for the study.
- “A significant stenosis was de- 104 fined by an artery diameter reduction ≥ 50 %” - although great confusion exists, a stenosis of 50% is usually considered intermediate, whereas a significant stenosis is usually identified when ≥ 70%. Please comment and clarify the choice.
- In case normality is not verified, one should specify the use of median (IQR).
- A half of eligible patients are excluded for some reason. The study is clearly at high risk of selection bias.
- Why did you perform a multivariate analysis knowing that no predictor had been found at univariate analysis? I would remove it.
- It would be useful to briefly describe the CAD-RADS score.
- I wouldn’t remark that CT scan is feasible. We don’t need to prove feasibility, we all use CT scan to assess CAD irrespectively of aortic stenosis. How good is CT scan in this setting is a different issue.
- The article would benefit from a careful proofreading, English language is seriously flawed.
Author Response
Reviewer 3
Thank you for your thoughtful review of this study. We have modified the manuscript in accordance with your comments.
- The retrospective and monocentric design, together with the small sample size, are major limitation.
- The author themselves acknowledge that there are meta-analyses (which means several studies) on the same subject. It is unclear what the study add to the literature.
Our study provides diagnostic performance data for each artery that are not reported in these meta-analyses. In addition, we believe that as the risk profile of patients changes over time, additional data on currently managed patients seems of interest.
- Why was ICA adjudicated by two cardiologist whereas coronary CT scan just by one radiologist? CT scan should be of main interest for the study.
For purely logistical reasons, only one radiologist could be assigned to read all the CCTAs. I fully agree that this can be a limitation. We mentioned it in the revised version of the manuscript. If requested we could involve a 2nd radiologist from another centre to review the CT.
- “A significant stenosis was de- 104 fined by an artery diameter reduction ≥ 50 %” - although great confusion exists, a stenosis of 50% is usually considered intermediate, whereas a significant stenosis is usually identified when ≥ 70%. Please comment and clarify the choice.
We defined coronary stenosis greater than 50% as significant by analogy to the CAD-RADS score. Indeed, the latter classifies lesions greater than 50% in category 3 justifying further exploration. In this particular population, we wished to propose a protocol that would allow us not to perform coronary angiography in case of a CAD-RADS score strictly inferior to 3. In addition, other studies on the subject also use a significance level of 50%. We now specify this in the revised version.
- In case normality is not verified, one should specify the use of median (IQR).
We added this information.
- A half of eligible patients are excluded for some reason. The study is clearly at high risk of selection bias.
We mentionned it in the discussion.
In our study, 58 eligible patients (17%) were excluded because of unavailable or incomplete CCTA, or PCI performed before CCTA. We think that this corresponds to real-world situations. However, other studies have had fewer excluded patients. Gohmann et al. (20) excluded only 7 of 517 patients (1.4%) due to incorrect CT-protocol; this could be explained by the prospective design of the study. In their retrospective study, Strong et al. (8) excluded 30 of 342 patients (8.8%); this could be explained by the fact that they included only patients who had CCTA available to them, as well as patients for whom PCI was performed before CCTA. They excluded patients for whom the time between CCTA and ICA was greater than 6 months.
- Why did you perform a multivariate analysis knowing that no predictor had been found at univariate analysis? I would remove it.
Indeed, you are absolutely right. This analysis is not related to the objective of our study which is not large enough to show a relevant result. To avoid any confusion we have removed this part of the manuscript.
- It would be useful to briefly describe the CAD-RADS score.
We now describe it in materials and methods. CAD-RADS 0: no stenosis; 1: 1-24% stenosis; 2: 25-49% stenosis; 3: 50-69% stenosis; 4: 70-99% stenosis or left main stem stenosis ≥50%; 5: totally occluded artery.
- I wouldn’t remark that CT scan is feasible. We don’t need to prove feasibility, we all use CT scan to assess CAD irrespectively of aortic stenosis. How good is CT scan in this setting is a different issue.
Your comment is very accurate. We have removed the sentences dealing with the feasibility of the CCTA. Indeed, our only objective is the analysis of the diagnostic performance of the CCTA.
- The article would benefit from a careful proofreading, English language is seriously flawed.
The revised version of the manuscript was proofread by English medical writer. You will find the final version of the revised paper.

Round 2
Reviewer 1 Report
Nothing to mention.
English has been improved.
Limitations are numerous, mentioned in correct manner.
Author Response
Thank you for your reviewing and the improvements of the manuscript we could made thanks to it.

Reviewer 3 Report
Thanks for the amendments. I have a few further observations:
- In my experience, it is very common to see patients coming to the cathlab with suspected disease (according to CT) and then found out to have normal coronary arteries. In keeping with this, I see from your study that false positive are very common. I would suggest that you provide further analyses to compare false and true positive, and if possible to understand what the predictors of a false positivity are. This should be also discussed in the manuscript.
- I would suggest to better discuss the clinical relevance of the CAD-RADS score. The reader should understand if and why it is meaningful to refer to this score.
- As you are arguing that it's important to repeat analyses over time to make them more contemporary, I would recommend discussing the differences between yours and other similar studies included in previous meta-analyses.
Author Response
Thank you for your thoughtful review of this study. We have modified the manuscript in accordance with your comments.
- In my experience, it is very common to see patients coming to the cathlab with suspected disease (according to CT) and then found out to have normal coronary arteries. In keeping with this, I see from your study that false positive are very common. I would suggest that you provide further analyses to compare false and true positive, and if possible to understand what the predictors of a false positivity are. This should be also discussed in the manuscript.
We didn’t compare true and false positive because of a limited sample size (43 true positive and 23 false positive) which didn’t allow us to highlight a statistically significant difference. Moreover, the final objective of our study was to confirm that by using CCTA we could safely exclude CAD and not confirm it. Guidelines tell us that CCTA must be used in case of low probability of CAD to exclude it. Litterature confirms that CCTA had a too low specificity and positive predictive value ; this shows us that CCTA is not good to confirm with a high degree of confidence presence of CAD.
- I would suggest to better discuss the clinical relevance of the CAD-RADS score. The reader should understand if and why it is meaningful to refer to this score.
We chose to use the CAD-RADS score to describe the coronary lesions demonstrated on CCTA because of its correlation with coronary angiography and its prognostic value. Indeed, Xie et al. (20) demonstrated in their registry including about 5000 patients that a higher CAD-RADS score was associated with a higher rate of cardiovascular events at 5 years. In the randomized controlled SCOT-HEART study (21), the use of CCTA and classification of lesions by CAD-RADS score in addition to standard care in patients with stable chest pain resulted in a significantly lower rate of death from coronary heart disease or non-fatal myocardial infarction at 5 years than standard care alone without resulting in a significantly higher rate of ICA.
Moreover, this score also remains the most widely used to describe lesions on CCTA, making it easier to extrapolate our results.
This is now discussed in the new version off the manuscript.
- As you are arguing that it's important to repeat analyses over time to make them more contemporary, I would recommend discussing the differences between yours and other similar studies included in previous meta-analyses.
It is important to complete the available data on the subject because of the technical improvements that can improve the diagnostic performance of the CCTA. In our study, we used a 256-row detector CT while in the available meta-analysis most studies used 64 and 126-row detector CT.
We mention it in the new version of the manuscript.
When analyzing the characteristics of the populations of other studies, there is no major difference in the patient profile, although in most of them the risk scores STS score or Euroscore are not mentioned.
